# Kinematic Alignment in Total Knee Arthroplasty Reduces Polyethylene Contact Pressure by Increasing the Contact Area, When Compared to Mechanical Alignment—A Finite Element Analysis

**DOI:** 10.3390/jpm12081285

**Published:** 2022-08-05

**Authors:** Antonio Klasan, Andreas Kapshammer, Veronika Miron, Zoltan Major

**Affiliations:** 1Johannes Kepler University, 4040 Linz, Austria; 2AUVA UKH Styria, 8020 Graz, Austria; 3Institute of Polymer Product Engineering, 4040 Linz, Austria

**Keywords:** total knee arthroplasty, kinematic alignment, mechanical alignment, UHMWPE, finite element analysis

## Abstract

Unrestricted Kinematic alignment (KA) in total knee arthroplasty (TKA) replicates the joint line of each patient by adjusting the cuts based on the anatomy of the patient. Mechanical alignment (MA) aims to restore a neutral mechanical axis of the leg, irrespective of the joint line orientation. The purpose of the present study was to compare contact pressure and contact areas of the polyethylene (PE) bearing surface as well as von Mises stress of the PE-tibial tray interface for MA and KA in the same patient, using CT data and finite element analysis. Finite element models were created from lower leg CT scans of 10 patients with knee osteoarthritis with different phenotypes. Mechanical PE properties were experimentally determined by tensile tests on dumbbell specimens. For numerical simulation purposes an adjusted non-linear material model with the maximum load to failure of 30.5 MPa, was calibrated and utilized. Contact pressure points were the deepest parts of the polyethylene inlay. Contact pressures were either very similar or were increased for MA knees throughout the gait cycle. KA either increased or had a comparable contact area, compared to MA. KA and MA produced comparable von Mises stresses, although both alignments breached the failure point of 30.5 MPa in all 3 valgus knees. This might indicate a higher probability of failure at the inlay-tibial baseplate interface. By maintaining the joint line orientation, KA reduces or has comparable contact pressures on the PE bearing surface by increasing or maintaining the contact area throughout one gait cycle in a validated finite element analysis model in 10 different knee phenotypes. The von Mises stress on the PE-tibial component interface was comparable, except for the valgus knees, where the load to failure was achieved in both alignment strategies and slightly higher stresses were observed for KA. Further studies for different knee phenotypes are needed to better understand the pressure changes depending on the alignment strategy applied.

## 1. Introduction

In total knee arthroplasty (TKA), unrestricted kinematic alignment (KA) has been proposed as an alternative to mechanical alignment (MA). Whereas MA aims to restore a neutral mechanical axis of the leg [1], KA aims to restore the native joint kinematics by aiming to restore the 3 kinematic axes of the knee [2]. Some of the concerns around KA were based on previous studies reporting increased loosening rates of the implants, in particular of the tibial component [3]. One of the pillars of MA was the fact that the implant position perpendicular to the mechanical axis equalizes the load on the medial and lateral aspects of the components, thus ensuring the longevity of the implant [1].

Mid- to long-term data did not demonstrate a difference in revision rate [4,5,6] and demonstrated comparable or slightly favorable clinical outcomes of KA over MA [5,7]. Regardless, some studies suggest that unrestricted KA can cause high loads on both the bone-implant interface as well as on the implants [8,9].

In earlier biomechanical tests, KA improved knee kinematics but demonstrated increased stress on the tibia [8]. Similarly, earlier finite element (FE) analyses found improved kinematics of KA over MA, but with increased contact stresses [9]. More recent FE studies have shown decreased stresses [10] or no difference [11].

Most previous studies revolve around a very small number of knee models or even one knee model, with a limited number of knee phenotypes, interpret KA in inconsistent ways, and use older or non-existing implants.

The purpose of this study was to compare polyethylene contact pressure, contact area and von Mises stresses in a cohort of 10 patients with different knee phenotypes.

## 2. Materials and Methods

### 2.1. Patient Data and Models

Data of 10 patients with knee osteoarthritis (OA) undergoing knee arthroplasty were used for the study, which was approved by our local ethical committee (Johannes Kepler University Linz, 1146/2021). CT scans of the hip, knee, and ankle were obtained using an established protocol used for partial and total knee arthroplasty [12]. Hip-knee-ankle angle (HKA), femoral mechanical angle (FMA), and tibial mechanical angle (TMA) were calculated for each patient [13]. Patient knee phenotypes were classified according to the classification proposed by Hirschmann et al. [14].

The transformation from CT data into a 3D model was done using 3D Slicer [15]. The models were edited and preprocessed using Siemens NX (Siemens AG, Munich, Germany) and Abaqus (Simulia Dassault Systems, Providence, RI, USA). The analyses were performed using the Abaqus Knee Simulator (Dassault Systems). The former computer-aided design (CAD) software provided the environment to perform preparations concerning the alignment strategies in a geometrical manner whereas the latter one was used to set up the numerical simulation including discretization, material models, boundary conditions, and loads. All analyses were performed using the CAD models of the 5C© total knee arthroplasty system (Implantcast, Buxtehude, Germany), using a cruciate retaining polyethylene with 9 mm thickness. Thus, the ligament properties also had to be considered within the numerical analysis. To realize this, the anatomically necessary ligaments were modeled by utilizing nonlinear, elastic connector properties to keep the calculational effort reasonable. Further, a hyperelastic material model describes the mechanical response of the shell elements discretized vastus intermedius, rectus femoris, and the patella ligament. The 5C© (Implantcast) has 12 femoral component sizes, with sizes 8–12 also in a narrow version. The component is posterior referenced, single-radius, bone sparing, with non-linear growth between 1.7 and 2.9 mm between sizes anterior-posterior. There are 12 tibial component sizes, with 7 inlay heights.

### 2.2. Implantation and Alignment

For MA TKA, CT-referenced digital implantation was performed by adjusting the femoral and the tibial cut perpendicular to the respective mechanical axes, by removing 7 mm from the most distal point of the femoral condyle and 7 mm from the deepest point of the prouder side of the tibial plateau. Assuming a native 1–2 mm cartilage thickness, this achieves the implant thickness of 8 and 9 mm respectively. The tibial component was implanted at 3° of the posterior slope, the femur was flexed at 6°, and rotated parallel to the trans-epicondylar axis. For KA TKA, 7 mm were removed in all planes with the implants oriented parallel to these cuts. The tibial component was implanted at 3° of the posterior slope, the femur was flexed at 6°. The femur was sized to avoid notching and mediolateral overhang. A dual fellowship-trained orthopedic knee surgeon and a biomechanical engineer performed the implant position and determined the sizes for all cases. In total 20 models were generated referring to KA and MA TKA for each analyzed patient.

### 2.3. Finite Element Analysis

In this study, the average gait cycle of a human being was analyzed according to a representation of the corresponding apparent forces and torques in the human knee, using a validated FE model [16]. Concerning the material modeling of the single components, the bones were assumed as rigid because the main focus was on the deformation and interaction of the single implant’s tibia tray, tibia insert, and femoral component. The first and third ones were modeled as steel using a linear elastic model with a Young’s modulus of E = 210,000 MPa and a Poisson ratio of v = 0.3. The tibial insert was modeled as a non-linear hybrid material model specially designed for ultra-high molecular weight polyethylene (UHMWPE). The model contains 18 parameters which were calibrated by experimentally carried out tensile tests on the UHMWPE grade Gur-4130, manufactured in compression molding applying a compression force of ten tons [17]. The material model is based on the work of Bergström et al. [18,19,20]. As a reference, the maximum engineering stress before failure was recorded at a value of σ_(eng,max) = 30.5 MPa.

After the finite element calculations had finished, the contact pressure and contact area of the UHMWPE insert were calculated. The measurement points were the medial and lateral deepest points on the UHMWPE insert, Figure 1. Finally, the maximum measured von Mises stress between the tibia insert and the tibia tray was investigated.

## 3. Results

Patient demographics and phenotypes are shown in Table 1.

As a result of evaluating the measurement points, Figure 2, with respect to the contact pressure, one can track the evolution of the parameter over a whole gait cycle. The generated line plots for all patients comparing medial/lateral and KA/MA are provided in Figure 3. The solid lines represent KA configurations, while the dashed one indicates MA configurations. To distinguish between the medial and lateral position of the investigated point blue and orange color is used, respectively.

The contact pressures were either very similar or were increased for MA knees throughout the gait cycle and have higher maximum values. In general, very volatile characteristics shown by the valgus knees of P2, P6, and P7 are even amplified by MA configurations, except briefly at the end of the cycle for P7. Coinciding with the force and torque plots of the boundary conditions, all configurations show maximum values at around 2.5 s. A maximum is reached in the case of P2–MA, a lateral point with a value of 28.2 MPa. Further, a trend of shifting the higher contact pressure to the lateral side can be observed.

With respect to the contact area between the tibial tray and the femoral component, KA either increased or had a comparable contact area, compared to MA, respectively, Figure 4.

There were some notable differences in von Mises stresses, Figure 5. In a few cases, especially valgus phenotype, P2, P6, and P7, both MA and KA breached the maximum stress of 30.5 MPa the UHMWPE can theoretically withstand, which is indicated by the dashed red line. A slight trend for higher stress concentrations in KA configurations is visible. Nevertheless, the finite element analyses of TKA in valgus knees, predict much higher stress concentrations compared to varus knees. In the right part of Figure 5, a typical example of such a stress concentration is shown. All maximum von Mises stress was located in the same region for all investigated configurations.

## 4. Discussion

This study demonstrates that KA reduces or has comparable contact pressures on the PE bearing surface at the beginning of the loading by increasing or maintaining the contact pressure throughout one gait cycle in a finite element analysis model. The maximum von Mises stress on the PE-tibial component interface was comparable between KA and MA, except for the valgus knee. However, the overall stress field in the PE shows in general a different distribution for KA and MA.

The best alignment of TKA is still a matter of debate. Until the emergence of KA, MA was regarded as the gold standard [1], and, according to some authors, still is [5]. The coronal malposition in MA has been shown to be a reason for failure [3]. This has been corroborated by finite element analyses. Stan et al. used a single FE model and implanted the tibia in 0°, 3°, and 8° varus. The authors find that imbalanced knees, those with increasing varus, increase the load on the medial tibia for a varus knee [21]. Balance is very difficult to assess using FE, even using conventional instruments, without digital tools [22]. Nowadays, coronal varus and valgus are not regarded as a malposition, one can only mention component position mismatch, creating, for instance, medial tightness, such as observed in Stan et al. Innocenti et al. [23] demonstrated that varus/valgus deviation all cause increased contact pressure and contact area when compared to balanced, or, neutral alignment. The implant used in the study was not commercially available. More importantly, the authors arbitrarily chose 2°,4°, and 6° of varus and valgus, which does not directly correspond to KA, which orients itself on the anatomy of the patient. Knee phenotyping [14] has demonstrated that the population has a much wider variety of alignments. Innocenti et al. also used only one knee model. These findings were recently corroborated by Tang et al. [24] using neutral alignment, 3°, 5°, and 7° of both varus and valgus of both components in all combinations.

Dong Song et al. [25] recently compared MA and KA strategies in a finite element analysis of a medial pivoting knee design. Although KA achieved closer-to-normal knee biomechanics by allowing more posterior translation of the lateral tibial plateau, the authors did find higher contact stresses in KA than in MA [25]. Compared to the present study, contact pressures breached 20 MPa in valgus knees and MA only. In KA knees, the pressures were also higher for valgus knees but never reached 20 MPa. The loads in valgus knees remain higher due to the altered joint line and load distribution. Restoration of the mechanical axis to 0° does not seem to reduce the pressure, however, compared to KA, increases it. Valgus knees require difficult balancing with excessive soft tissue releases, which is why some authors propose a more restricted approach to KA, especially valgus knees [26]. The number of soft tissue releases needed to reduce the pressure is difficult to quantify in an FE analysis, but the lower pressure observed in the present study needs to be noted.

Earlier studies contradict our findings. Nakamura et al. [8] found the opposite effect for varus knees. In a severe varus case, the load medially in KA was increased between 32.2 and 53.7%. Since a soft-tissue release is difficult to perform in an FE analysis, these results contradict our findings. Ishikawa et al. [9] observed better kinematics but higher pressures, but the KA implantation was not model specific, but rather determined based on previous studies at 3° valgus for the femur and 3° varus for the tibia, which is anatomical alignment, not KA, since KA solely depends on the patient anatomy.

Recent studies corroborate our findings, which might be due to better models, but also due to a better understanding of KA philosophy by respecting the patient-specific anatomy. Kang et al. [10] investigated a cruciate retaining implant in a single patient and found improved kinematics of KA over MA, with reduced pressure due to the increased area, as observed in the present study. Von Mises stresses observed in the present study might indicate the reason for loosening in some valgus knees, since the stresses are increased on the polytheylene metal interface. This might occur as a consequence of an increased correction needed for valgus knees, which was, interestingly, observed in both alignment philosophies. A potential solution for excessive corrections might be a lateral pivoting knee since a dual-pivot phenomenon has been recently observed [27]. Another viable option to mitigate any issues is higher constraint [28].

Some limitations and assumptions need to be noted. The bone models were assumed to be rigid, rather than linear elastic. However, the polyethylene and metal have been programmed to behave based on validated and real-time tested data. Although a variety of phenotypes have been used in the study, the definitive pressures also depend on soft tissue releases, which are difficult to quantify in a FE analysis. The collateral ligaments observe significant strains in the native knee, which then change further during OA, with marked differences for each collateral ligament, depending on the varus/valgus phenotype [29,30]. Biomechanical validation of the findings, although interesting, is impossible since a single knee cannot be implanted with an MA and a KA philosophy. Two same phenotypes might have different implant sizes or previous ligament strains, which again, limits the viability of biomechanical testing.

## 5. Conclusions

By maintaining the joint line orientation, depending on the knee phenotype, KA reduces or has comparable contact pressures on the PE bearing surface by increasing or maintaining the contact area throughout one gait cycle in a validated finite element analysis model in 10 different knee phenotypes. The von Mises stress on the PE-tibial component interface was comparable, except for the valgus knees, where the load to failure was achieved in both alignment strategies and slightly higher stresses were observed for KA. Further studies for different knee phenotypes are needed to better understand the pressure changes depending on the alignment strategy applied.

## Figures and Tables

**Figure 1 jpm-12-01285-f001:**
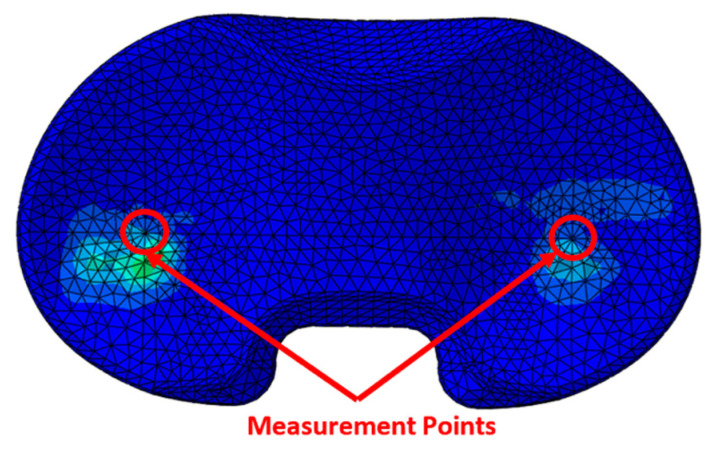
Measurements points concerning the extraction of the contact pressure values.

**Figure 2 jpm-12-01285-f002:**
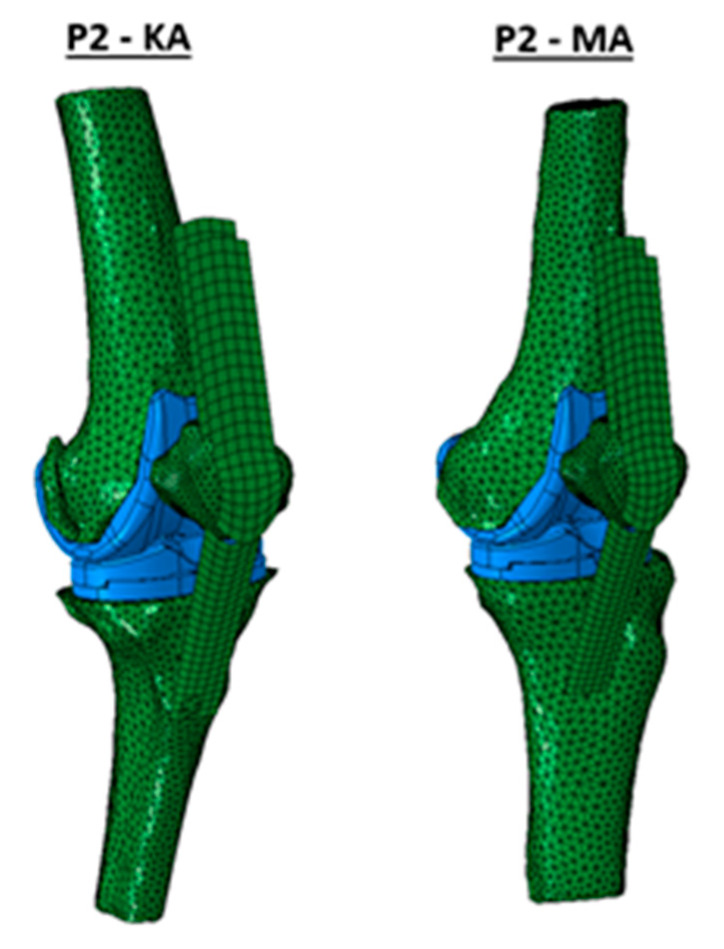
Illustration of the KA and MA configuration, in the finite element analysis, for the second used data set.

**Figure 3 jpm-12-01285-f003:**
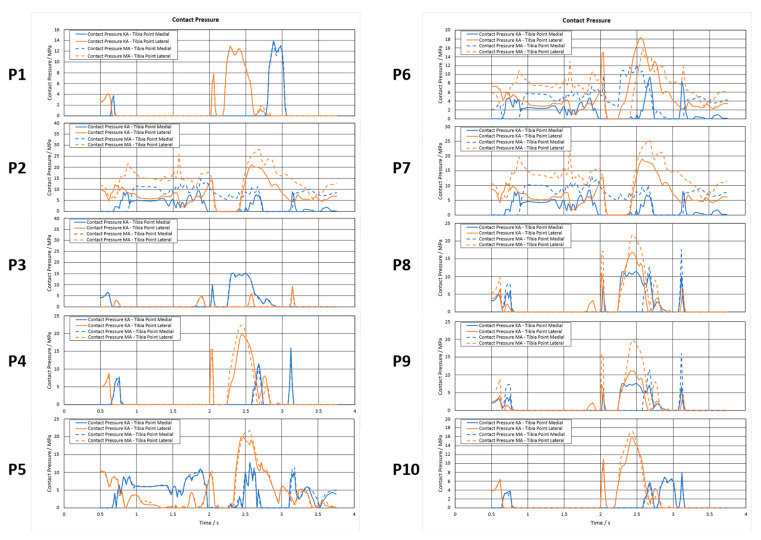
Contact Pressure evolution in MPa over one gait cycle, comparing medial/lateral points and KA/MA.

**Figure 4 jpm-12-01285-f004:**
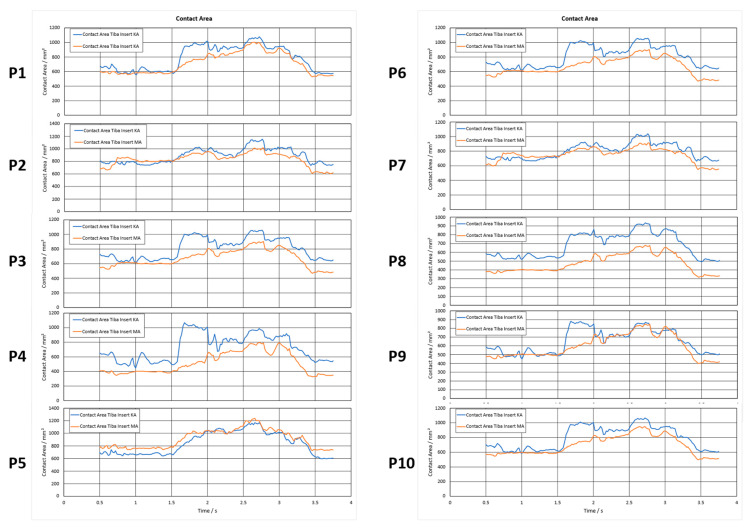
Contact area evolution in mm² of the tibia inserts over one gait cycle.

**Figure 5 jpm-12-01285-f005:**
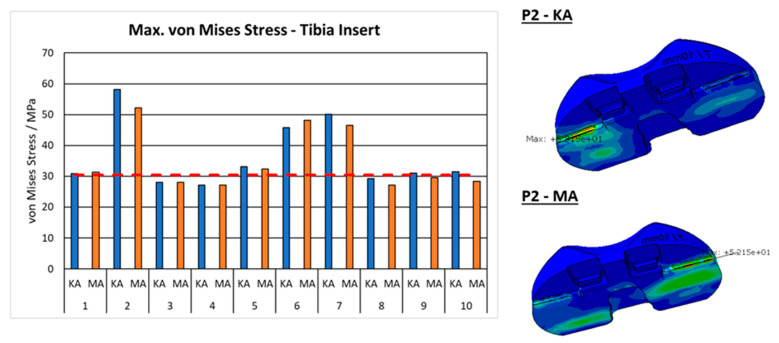
Maximum resulting von Mises stress considering the whole tibia insert. An example of P2 where the maximum occurs.

**Table 1 jpm-12-01285-t001:** Hip-Knee-Ankle (HKA) angle configurations of the analyzed knee joints.

Patient	Age (Years)	Gender	Knee Phenotype
P1	71	M	VAR(HKA)3, VAR(FMA)3, VAL(TMA)3
P2	73	F	VAL(HKA)9, VAL(FMA)6, VAL(TMA)6
P3	59	M	VAR(HKA)9, NEU(FMA)0, VAR(TMA)3
P4	56	F	VAR(HKA)3, NEU(FMA)0, VAR(TMA)3
P5	62	M	VAR(HKA)9, VAR(FMA)3, VAR(TMA)3
P6	57	M	VAL(HKA)6, NEU(FMA)0, VAL(TMA)6
P7	84	M	VAL(HKA)9, VAL(FMA)3, VAL(TMA)6
P8	75	F	VAR(HKA)6, NEU(FMA)0, NEU(TMA)0
P9	82	F	VAR(HKA)6, NEU(FMA)0, NEU(TMA)0
P10	83	M	VAR(HKA)9, VAR(FMA)6, VAR(TMA)9

## Data Availability

Data is not publicly available.

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
