# Peer review of "Kinematic Alignment in Total Knee Arthroplasty Reduces Polyethylene Contact Pressure by Increasing the Contact Area, When Compared to Mechanical Alignment—A Finite Element Analysis"

_jpm, 2022, doi:10.3390/jpm12081285_

Round 1
Reviewer 1 Report
This manuscript utilized FEM to analyze the stress behavior of kinematic alignment and mechanical alignment in total knee arthroplasty. Some comments are given as follows:
1. The title is too long and confusing. It should be a precise keynote, rather than a long sentence.
2. The FEM model description is missing lots of specific information, e.g., mesh size, boundary condition, and loading profile. It's hard to tell if the model is set up correctly, as there's no validation tested shown in the manuscript.
3. How is the model shown in Fig. 1 connected to the model in Fig.2?
4. Fig. 3 is blurred, hard to see the axis and legend labels, please replot.
5. Many sentences are confusing, e.g., line 20-21, 24-26
6. Is there any real medical results to show the advantage of KA compared with MA as claimed in this manuscript through a numerical study?
7. The reference number should be given right after the mentioned author, rather than at the end of the sentence. For instance, in line 200-202, it should be 'Kang et al. [10] investigated a cruciate retaining implant in a single patient and found 200 improved kinematics of KA over MA, with reduced pressure due to the increased area, as 201 observed in the present study .'
Author Response
Please s. the attached file.

Reviewer 2 Report
Lines 75-83: Where did the authors base their modeling assumptions? Are there any relevant references? They are kindly requested to elaborate more on this topic. Additionally, computational effort is mentioned as a factor in choosing elastic properties for the ligament material. Is this something that is widely acceptable? Some references should be provided here as well. Finally, some mention of element size or number should be provided to offer information on mesh density.
Lines 99-100: The authors are kindly requested to provide some additional information on what they refer to as a validated FE model. To the reviewer’s understanding, this refers to the application of loads and/or boundary conditions and the type of analysis (explicit/implicit) employed to simulate the gait cycle. More details on these would enhance the value of the text and provide the reader with necessary information to better understand the discussion that follows.
Lines 102-104: The authors are kindly requested to clarify if this assumption is supported by the validated FE model mentioned previously in the text (Ref. [16]).
Lines 98-114: Throughout the FEA section, the authors have failed to provide any substantial information on their meshing strategy, the order of the shell elements employed and if any convergence analysis was applied for their purposes. They have also failed to communicate how the interactions between the insert and bones were simulated. This information should be conveyed in this text as well, even if the validated model of Ref. 16 indeed covers all these topics.
Lines 118-119: The meaning of this sentence is unclear. The authors are kindly requested to rephrase, and consequently justify the value of adding Figure 2 to the text.
Figure 3 & 4: The authors are kindly requested to provide higher quality figures, since at this version they are hard to read, even when zoomed in.
Lines 143-145: Although stating that the von Mises stress exceeds the maximum acceptable values, the authors do not provide any further discussion on this topic. From an engineering perspective, this is unacceptable. Therefore, they are kindly requested to elaborate on this topic further and offer:
a) Justification on why this occurs from a physical standpoint and whether this is considered as acceptable under any circumstances.
b) Discussion on what this could entail for the structural integrity of the implant.
c) Suggestions on what could mitigate this problem for patients with this distinctive knee phenotype.
General Comments: Overall, this work offers an exposition of results obtained from performing a high-fidelity structural analysis on a complex biomechanical problem, for an expanded dataset compared to previous cases. However, other than that the authors do not provide any strong arguments on the scientific merit of their work. Although they state some interesting conclusions, e.g., that maximum stress exceeds acceptable limits for a certain category of knee phenotypes, their discussion on them is limited. This is a general theme of this work, where detailed explanations are avoided in several topics, such as the aforementioned as well as the FE modeling. It should be expected from the authors to not only provide discussion on such results, but also to be motivated by them to seek mitigating solutions. These are key aspects missing from this manuscript, which to the reviewer’s opinion should be addressed, if it is to be accepted for publication.
Author Response
Please s. the attached file.

Round 2
Reviewer 1 Report
The authors barely answer and clarify the concerns raised by the reviewer.
The authors claimed the FEM study has been detailed in the reference and refused to give any specifics in this manuscript, and also claimed there's no data for the validation.
Then the ultimate question is: what are the novelty and scientific soundness for this paper?
It is not suggested to get this manuscript published until the reviewer's comments have been properly addressed.
Reviewer 2 Report
Publish as is